# NEURON-ENHANCED AUTOENCODER BASED COLLABORATIVE FILTERING: THEORY AND PRACTICE

## ABSTRACT

This paper presents a novel recommendation method called neuron-enhanced autoencoder based collaborative filtering (NE-AECF). The method uses an additional neural network to enhance the reconstruction capability of autoencoder. Different from the main neural network implemented in a layer-wise manner, the additional neural network is implemented in an element-wise manner. They are trained simultaneously to construct an enhanced autoencoder of which the activation function in the output layer is learned adaptively to approximate possibly complicated response functions in real data. We provide theoretical analysis for NE-AECF to investigate the generalization ability of autoencoder and deep learning in collaborative filtering. We prove that the element-wise neural network is able to reduce the upper bound of the prediction error for the unknown ratings, the data sparsity is not problematic but useful, and the prediction performance is closely related to the difference between the number of users and the number of items. Numerical results show that our NE-AECF has promising performance on a few benchmark datasets.

## 1 INTRODUCTION

Recommendation system aims to provide personalized recommendation based on various information such as user purchase records, social networks, user features, and item (or product) features. With the fast growth of E-commence, social media, and content provider, recommendation systems play more and more important roles in our daily life and have changed our life both explicitly and implicitly. In general, recommendation systems can be organized into three categories (Adomavicius & Tuzhilin, 2005; Zhang et al., 2019): content based method, collaborative filtering, and hybrid methods. The content based methods recommend similar items to a user or recommend one item to similar users, where the similarity is usually obtained from side information such as genre, occupation, and age. Collaborative filtering (CF) assumes that there exist potential correlations within both users and items, which can be implicitly used to predict unknown ratings. Hybrid methods are combinations of content based methods and CF methods (Adomavicius & Tuzhilin, 2005; Zhang et al., 2019; Su & Khoshgoftaar, 2009). CF is at the cores of many recommendation systems.

Early CF methods (Resnick et al., 1994) compute the similarity between users or items directly from the ratings to make prediction. This kind of method is also called memory based CF, which is easy to implement and has high interpretability. One limitation is that the similarity computed from the ratings is not informative owing to the high sparsity of the rating. Another line of CF is model based method (Ungar & Foster, 1998; Shani et al., 2002) that utilizes historical data to train a machine learning model such as Bayesian network (Breese et al., 1998; Miyahara et al., 2000) for recommendation. Model based methods are more effective than content based methods in learning complex hidden preference and handling the sparsity problem. Note that both content based and model based methods do not work when there are new items or users without ratings, which is known as the cold start problem. A popular strategy for solving the problem is to incorporate side information into CF methods (Adams et al., 2010; Welling et al., 2012; Zhang et al., 2017).

In the past decades, matrix factorization (Billsus & Pazzani, 1998; Mnih & Salakhutdinov, 2008; Koren et al., 2009) and matrix completion (Candès & Recht, 2009; Shamir & Shalev-Shwartz, 2014; Sun & Luo, 2015; Chen et al., 2016; Fan et al., 2019) have been extensively studied and used in CF. These methods usually exploit the potential low-rank structure of the incomplete rating matrix

via embedding items and users into a latent space of reduced dimension, where the observed ratings are approximated by the inner products of the user feature vectors and item feature vectors. The low-rankness is usually obtained by low-rank factorization (Koren et al., 2009), nuclear norm minimization (Candès & Recht, 2009), or Schatten-$p$ quasi norm minimization (Fan et al., 2019). Particularly, Lee et al. (2016) proposed a local low-rank matrix approximation (LLORMA) that approximates the rating matrix as a weighted sum of a few low-rank matrices. LLORMA outperformed vanilla low-rank matrix completion methods in collaborative filtering, which indicates that the rating matrices in real applications may have more complicated structures rather than a single low-rank structure.

The success of neural networks and deep learning in computer vision and natural language processing inspired researchers to design neural networks for CF (Salakhutdinov et al., 2007; Dziugaite & Roy, 2015; Sedhain et al., 2015; Wu et al., 2016; Zheng et al., 2016; He et al., 2017; van den Berg et al., 2017; Fan & Cheng, 2018; Yi et al., 2020). For instance, Salakhutdinov et al. (2007) proposed a restricted Boltzmann machines (Hinton et al., 2006) based CF method called RBM-CF, which showed high performance in the Netflix challenge (Bennett & Lanning, 2007). Sedhain et al. (2015) proposed AutoRec, an autoencoder (Hinton & Salakhutdinov, 2006; Bengio et al., 2007) based CF method, which predicts unknown ratings by an encoder-decoder model $\hat{\boldsymbol{x}} = \boldsymbol{W}_2\sigma(\boldsymbol{W}_1\boldsymbol{x})$, where $\boldsymbol{x}$ denotes the incomplete ratings on one item or of one user and $\boldsymbol{W}_1, \boldsymbol{W}_2$ are weight matrices to optimize. Unlike RBM-CF, which is probabilistic and generative, AutoRec provides a discriminative approach. AutoRec outperformed LLORMA slightly on several benchmark datasets (Sedhain et al., 2015). In addition, adding depth is able to improve the performance of AutoRec (Sedhain et al., 2015). Inspried by Neural Autoregressive Distribution Estimator (NADE) (Larochelle & Murray, 2011) and RBM-CF (Salakhutdinov et al., 2007), Zheng et al. (2016) proposed a method called CF-NADE, in which parameters are shared between different ratings and achieved promising performance in several benchmarks. Muller et al. (2018) proposed a kernel based reparametrized neural network, in which the weight between two units is set to be a weighted kernel-function of the location vectors. The method works well in data visualization and recommendation systems. Interestingly, Yi et al. (2020) found that the expected value of the output layer of a neural network depends on the sparsity of the input data. They proposed a simple yet effective method called sparsity normalization to improve the performance of neural networks with sparse input data such as the highly incomplete rating matrices in CF.

It is worth mentioning that existing autoencoder based CF methods such as (Sedhain et al., 2015; Wu et al., 2016; Muller et al., 2018; Yi et al., 2020) use linear activation function in the output of the decoder, i.e., $\hat{\boldsymbol{x}} = \boldsymbol{W}_L\boldsymbol{h}_{L-1}$, where $\boldsymbol{W}_L$ denotes the weights of the output layer and $\boldsymbol{h}_{L-1}$ denotes the features given by the last hidden layer. Thus, these methods are under the assumption that the ratings are linear interactions between user features and item features, though the features can be nonlinear. Such an assumption may not be true or not optimal in real problems, especially when the data are bounded (e.g. images) or are collected by sensors (e.g. medical and chemical sensors) with nonlinear response functions. We suspect that the rating values given by users on items are from some nonlinear response functions because humans have complex emotion or decision curves (LeDoux, 2000; Baker, 2001). A naive method to incorporate nonlinear interaction is using nonlinear activation functions such as sigmoid function (with rescaling) in the output layer of the decoder, which however has much lower performance than using a linear activation function. That's why existing autoencoder based CF methods use only linear activation function. Note that a pre-specified activation function for the output layer of the decoder may work on specific data but may be far away from the possible optimal choice. On the other hand, the theoretical analysis for autoencoder and deep learning based CF is very limited, while there have been many works on the theory of low-rank matrix completion based CF (Srebro & Shraibman, 2005; Candès & Recht, 2009; Shamir & Shalev-Shwartz, 2014; Fan et al., 2019).

**Contribution.** In this paper, we present a novel neural network CF method named NE-AECF, an enhanced autoencoder approach for recommendation system. NE-AECF is composed of two different neural networks, one is an autoencoder to reconstruct the incomplete rating matrix, while the other is an element-wise neural network to learn an activation function adaptively for the output layer of the autoencoder. We provide theoretical analysis for NE-AECF, which explains the superiority of our method. Specifically, we prove that the element-wise neural network can reduce the upper bound of the prediction error for the unknown ratings. We also prove that the data sparsity is not problematic but useful and the prediction performance is closely related to the difference between the number of

users and the number of items. Further, we demonstrate empirically our NE-AECF on benchmarks: MovieLen-100k and MovieLen-1M, achieving state-of-the-art results.

**Notation.** We use $x$ (or $X$), $\boldsymbol{x}$, and $\boldsymbol{X}$ to denote scalar, vector, and matrix respectively. We use $\|\boldsymbol{x}\|$ to denote the Euclidean norm of vector $\boldsymbol{x}$, use $\|\boldsymbol{X}\|_F$ and $\|\boldsymbol{X}\|_2$ to denote the Frobenius norm and spectral norm of matrix $\boldsymbol{X}$ respectively. The $\ell_{21}$ norm of matrix is denoted by $\|\boldsymbol{X}\|_{2,1} := \sum_i \|\boldsymbol{x}_i\|$, where $\boldsymbol{x}_i$ denotes the $i$-th column of $\boldsymbol{X}$. The $\ell_\infty$ norm of matrix is denoted by $\|\boldsymbol{X}\|_\infty := \max_{ij} |X_{ij}|$. We use $|S|$ to denote the cardinality of set $S$. The symbol $'\odot'$ denotes the Hadamard product between vectors or matrices. The symbol $'\circ'$ denotes function composition.

## 2 Neuron-Enhanced AECF

Suppose we have an incomplete rating matrix $\tilde{\boldsymbol{X}} = (\tilde{\boldsymbol{x}}_1, \tilde{\boldsymbol{x}}_2, \ldots, \tilde{\boldsymbol{x}}_n) \in \mathbb{R}^{m \times n}$, where $m$ is the number of users and $n$ is the number of items (without loss of generality). $\tilde{X}_{ij} \geq 0$ denotes the rating given by user $i$ on item $j$ and $\tilde{X}_{ij} = 0$ indicates an unobserved rating. $S$ denotes the set of observed ratings. We have $\tilde{X}_{ij} = X_{ij}$ for all $(i, j) \in S$. Our goal is to predict the unobserved ratings $X_{ij}, (i, j) \in [m] \times [n] \backslash S$, from $\tilde{\boldsymbol{X}}$.

We want to learn a nonlinear function $f : \mathbb{R}^m \mapsto \mathbb{R}^m$ such that

$$\sum_{i=1}^n \left\| \boldsymbol{s}_i \odot \left( \tilde{\boldsymbol{x}}_i - f(\tilde{\boldsymbol{x}}_i) \right) \right\|^2 \tag{1}$$

is as small as possible, where $\boldsymbol{s}_i$ is a binary vector denoting whether the the corresponding element in $\boldsymbol{x}_i$ is zero (unknown) or not. The motivation is predicting the missing entries of $\boldsymbol{x}_i$ using its observed entries, though the missing entries of $\boldsymbol{x}_i$ are filled by zeros before performing $f$. More formally, we consider the following problem

$$\underset{f \in \mathcal{F}}{\text{minimize}} \left\| \boldsymbol{S} \odot \left( \tilde{\boldsymbol{X}} - f(\tilde{\boldsymbol{X}}) \right) \right\|_F^2 \tag{2}$$

where $\boldsymbol{S} = (\boldsymbol{s}_1, \boldsymbol{s}_2, \ldots, \boldsymbol{s}_n)$, $f$ is performed on each column of $\tilde{\boldsymbol{X}}$ separately, and $\mathcal{F}$ denotes a hypothesis set of $m$ to $m$ functions. We have infinite choices for $\mathcal{F}$. For example, $\mathcal{F}$ can be a set of functions in the form of neural network with some parameters $W \in \mathcal{W}$, where $\mathcal{W}$ denotes a set of matrices under some constraints. In this case, problem (2) defines a denoising autoencoder or stacked denoising autoencoders (Vincent et al., 2010), where the noises are introduced by filling the missing ratings with zeros.

Let $f$ be an autoencoder with linear activation function in the output layer. Then (2) becomes

$$\underset{\boldsymbol{W}_1, \boldsymbol{W}_2}{\text{minimize}} \left\| \boldsymbol{S} \odot \left( \tilde{\boldsymbol{X}} - \boldsymbol{W}_2 \sigma(\boldsymbol{W}_1 \tilde{\boldsymbol{X}}) \right) \right\|_F^2 + \lambda \left( \|\boldsymbol{W}_1\|_F^2 + \|\boldsymbol{W}_2\|_F^2 \right), \tag{3}$$

where $\boldsymbol{W}_1 \in \mathbb{R}^{d \times m}$ and $\boldsymbol{W}_2 \in \mathbb{R}^{m \times d}$ are weights matrices to learn and $\lambda$ is a nonnegative constant to control the strength of weight decay. We have omitted the bia terms for simplicity. $\sigma$ denotes an activation function such as

$$\text{ReLU } \sigma(x) = \max(x, 0) \text{ and}$$
$$\text{Sigmoid } \sigma(x) = 1/(1 + \exp(-x)).$$

Note that (3) is exactly the basic model considered by Sedhain et al. (2015),Wu et al. (2016),Muller et al. (2018), and Yi et al. (2020). Once (3) is used, the following assumption is made implicitly.

**Assumption 1.** *There exist two matrices $\boldsymbol{A} \in \mathbb{R}^{m \times d}$ and $\boldsymbol{B} \in \mathbb{R}^{d \times n}$ such that $\|\boldsymbol{S} \odot (\boldsymbol{X} - \boldsymbol{A}\boldsymbol{B})\|_F$ is small enough.*

The assumption indicates that if $d$ is much smaller than $\min(m, n)$, $\boldsymbol{X}$ can be well approximated by a low-rank matrix, which however may not always hold in real applications. Consider the following data generating model

$$\boldsymbol{X} = h(\boldsymbol{A}'\boldsymbol{B}'), \tag{4}$$

where $h : \mathbb{R}^1 \mapsto \mathbb{R}^1$ is an element-wise nonlinear function and $\boldsymbol{A}' \in \mathbb{R}^{m \times d}$, $\boldsymbol{B}' \in \mathbb{R}^{d \times n}$ may be generated by some nonlinear functions. If the nonlinearity of $h$ is high, $\boldsymbol{X}$ cannot be well approximated by a rank-$d$ matrix. These analysis indicates that if the element-wise nonlinearity in generating $\boldsymbol{X}$ is strong, (3) should use a large $d$ to ensure a small enough training error.

The element-wise nonlinearity widely exists in real data. For example, in imaging science, the intensity of pixels are nonlinear responses of photoelectric element to spectrum. In chemical engineering, many sensors have nonlinear responses. In biomedical engineering, the dose-responses are often nonlinear curves. Hence, in collaborative filtering, the ratings may be nonlinear responses to some latent values, according to the studies on response curve in neuroscience and psychology (LeDoux, 2000; Baker, 2001).

Therefore, instead of (3), one may consider the following problem

$$\underset{\boldsymbol{W}_1, \boldsymbol{W}_2}{\text{minimize}} \left\| \boldsymbol{S} \odot \left( \tilde{\boldsymbol{X}} - h\big(\boldsymbol{W}_2 \sigma(\boldsymbol{W}_1 \tilde{\boldsymbol{X}})\big) \right) \right\|_F^2 + \lambda \left( \|\boldsymbol{W}_1\|_F^2 + \|\boldsymbol{W}_2\|_F^2 \right), \tag{5}$$

where $h$ should be determined beforehand. A naive approach to determining $h$ is choosing a bounded or partially bounded nonlinear function according to the range of the data. For example, if the data are image pixels within $[0, 1]$, one may use Sigmoid function. If the data are nonnegative, one may use ReLU. However, such choices only considered the range of the data, which is just a small portion of the nonlinearity. Within the range, the true response functions are not necessarily linear (ReLU) or related to exponential (Sigmoid), and can be much more complicated.

As it is difficult to choose a suitable nonlinear function $h$ in advance, we propose to learn $h$ from the data adaptively, i.e.,

$$\underset{\boldsymbol{W}_1, \boldsymbol{W}_2, h \in \mathcal{H}}{\text{minimize}} \left\| \boldsymbol{S} \odot \left( \tilde{\boldsymbol{X}} - h\big(\boldsymbol{W}_2 \sigma(\boldsymbol{W}_1 \tilde{\boldsymbol{X}})\big) \right) \right\|_F^2 + \lambda \left( \|\boldsymbol{W}_1\|_F^2 + \|\boldsymbol{W}_2\|_F^2 \right), \tag{6}$$

where $\mathcal{H}$ denotes a hypothesis set of nonlinear functions from $\mathbb{R}^1$ to $\mathbb{R}^1$. We have different approaches to learning $h$. The first approach is combining various activation functions, i.e.,

$$h_\theta(z) = \sum_i^k \theta_i \sigma_i(z), \tag{7}$$

where $\sigma_i(\cdot)$ are different activation functions and $\boldsymbol{\theta} = (\theta_1, \ldots, \theta_k)^\top$ are parameters to estimate. However, it is not clear whether (7) is able to approximate a wide range of nonlinear functions. The second approach is using polynomial functions, i.e.,

$$h_\theta(z) = \sum_i^k \theta z^k. \tag{8}$$

It is a $k$-order polynomial function and can well approximate any smooth functions provided that $k$ is sufficiently large. Another approach is using a neural network, i.e.,

$$h_\Theta(z) = \boldsymbol{\Theta}_{L_\Theta}(\sigma_\Theta(\boldsymbol{\Theta}_{L_\Theta - 1} \sigma_\Theta(\cdots \sigma_\Theta(\boldsymbol{\Theta}_1 z) \cdots))), \tag{9}$$

where $\boldsymbol{\Theta}_1$ and $\boldsymbol{\Theta}_{L_\Theta}$ are vectors, $\boldsymbol{\Theta}_2, \ldots, \boldsymbol{\Theta}_{L_\Theta - 1}$ are matrices, and $\sigma_\Theta$ is a fixed activation function. According to the universal approximation theorems (Pinkus, 1999; Sonoda & Murata, 2017; Lu et al., 2017), (9) is able to approximate any continuous functions provided that the network is wide enough or deep enough.

Since (9) is more flexible than (7) and (8) in function approximation, we propose to solve the following problem

$$\underset{W, \Theta}{\text{minimize}} \left\| \boldsymbol{S} \odot \left( \tilde{\boldsymbol{X}} - h_\Theta\big(g_W(\tilde{\boldsymbol{X}})\big) \right) \right\|_F^2 + \lambda_W \sum_{l=1}^{L_W} \|\boldsymbol{W}_l\|_F^2 + \lambda_\Theta \sum_{l=1}^{L_\Theta} \|\boldsymbol{\Theta}_l\|_F^2, \tag{10}$$

where $W = \{\boldsymbol{W}_1, \ldots, \boldsymbol{W}_{L_W}\}$, $\Theta = \{\boldsymbol{\Theta}_1, \ldots, \boldsymbol{\Theta}_{L_\Theta}\}$, and

$$g_W(\tilde{\boldsymbol{X}}) = \boldsymbol{W}_{L_W}\left( \sigma_W\big(\boldsymbol{W}_{L_W - 1} \sigma_W(\cdots \sigma_W(\boldsymbol{W}_1 \tilde{\boldsymbol{X}}) \cdots)\big) \right). \tag{11}$$

In addition, we assume $\boldsymbol{W}_l \in \mathbb{R}^{d_l \times d_{l-1}}$, $l \in [L_W]$, and $\boldsymbol{\Theta}_l \in \mathbb{R}^{p_l \times p_{l-1}}$, $l \in [L_\Theta]$. Note that $d_0 = d_{L_W} = m$ and $p_0 = p_{L_\Theta} = 1$. Comparing (10) with (2), we see that we have replaced $f$ by $h_W \circ g_\Theta$ with Frobenius-norm constrained weight matrices. Model (10) is exactly our *neuron-enhanced autoencoder based collaborative filtering* (NE-AECF) method. There are two different neural networks. The first one is an autoencoder defined by $h_\Theta \circ g_W$, which is to learn an contraction map from the incomplete rating matrix $\tilde{\boldsymbol{X}}$ to itself or its observed entries more precisely. The second neural network is performed in an element-wise manner to learn an activation function $h$ adaptively for the output layer of the autoencoder or stancked autoencoders. Figure 1 shows an example schematic of NE-AECF, where $L_W = L_\Theta = 2$ and $\boldsymbol{Z} = g_W(\tilde{\boldsymbol{X}})$.

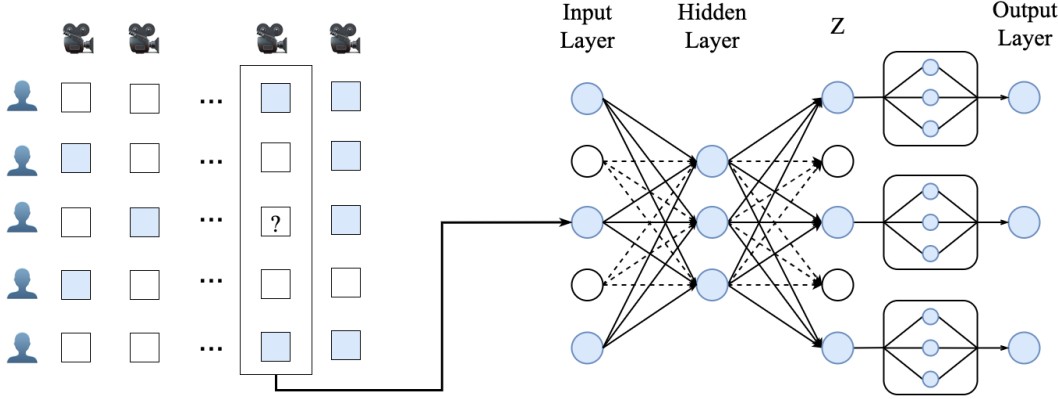

Figure 1: An example schematic of NE-AECF. The left part demonstrates a rating matrix where users and items are represented by each row and column. Every square in the left part corresponds to a rating. Observed ratings are colored and unobserved rating are left white. Target rating being predicted is marked with question mark.

## 3 GENERALIZATION ERROR BOUND OF NE-AECF

In this section, we analyze the capability of NE-AECF in predicting the unknown ratings of $\boldsymbol{X}$. Note that $\frac{1}{|S|}\|\boldsymbol{S} \odot (\boldsymbol{X} - \hat{\boldsymbol{X}})\|_F^2 = \frac{1}{|S|}\|\boldsymbol{S} \odot (\tilde{\boldsymbol{X}} - \hat{\boldsymbol{X}})\|_F^2 := \mathcal{L}_S$, where $\hat{\boldsymbol{X}} = h_\Theta(g_W(\tilde{\boldsymbol{X}}))$. We have

$$\mathcal{L}_S = \frac{1}{|S|} \sum_{(i,j) \in S} \ell(X_{ij}, \hat{X}_{ij}),$$

where $\ell(X_{ij}, \hat{X}_{ij}) = (X_{ij} - \hat{X}_{ij})^2$. Note that instead of the square loss, we may use other functions such as $|X_{ij} - \hat{X}_{ij}|$. In the remainder of this paper, $\ell(X_{ij}, \hat{X}_{ij})$ denotes a general loss. Let $S^c \triangleq [m] \times [n] \backslash S$, the generalization error of NE-AECF is quantified by

$$\mathcal{L}_{S^c} = \frac{1}{|S^c|} \sum_{(i,j) \in S^c} \ell(X_{ij}, \hat{X}_{ij}),$$

which is a measurement of the prediction error of NE-AECF for the unknown ratings in $\boldsymbol{X}$. We have the following generalization bound.

**Theorem 1.** *Suppose a set $S$ of ratings of $\boldsymbol{X} \in \mathbb{R}^{m \times n}$ are observed uniformaly and randomly, which results in an incomplete matrix $\tilde{\boldsymbol{X}}$ with unknown ratings replaced by some values such as zero. Suppose $mn - |S| > |S| > 50$. Let $\hat{\boldsymbol{X}} = h_\Theta(g_W(\tilde{\boldsymbol{X}}))$, where $h_\Theta$ is defined by (9) and $g_W$ is defined by (11). Suppose $\|\boldsymbol{W}_l\|_2 \leq a_l$, $\|\boldsymbol{W}_l\|_{2,1} \leq a'_l$, $l \in [L_W]$, $\bar{d} := \max(d_1, \ldots, d_{L_W-1}) < m$, and $\|\boldsymbol{\Theta}_l\|_2 \leq b_l$, $\|\boldsymbol{\Theta}_l\|_F \leq b'_l$, $l \in [L_\Theta]$. Suppose the Lipschitz constants of $\sigma_W$ and $\sigma_\Theta$ are $\rho$ and $\varrho$ respectively. Suppose $\sup_{i,j} |\ell(Y_{ij}, X_{ij})| \leq \tau_\ell$, $\ell$ is $\eta_\ell$-Lipschitz, and $\|\hat{\boldsymbol{X}}\|_\infty \leq \mu$. Then with*

*probability at least $1 - \delta$ over the random sampling $S$,*

$$
\frac{1}{|S^c|} \sum_{(i,j) \in S^c} \ell\left(X_{ij}, \hat{X}_{ij}\right) - \frac{1}{|S|} \sum_{(i,j) \in S} \ell\left(X_{ij}, \hat{X}_{ij}\right)
$$
$$
\leq \frac{C_1 \eta_\ell v_1 \ln |S|}{|S|} + C_2 \eta_\ell \mu \sqrt{\frac{v_2 \ln v_3}{|S|}} + \frac{11\tau_\ell mn}{\sqrt{|S^c|}|S|} + 3\tau_\ell \sqrt{\frac{mn}{|S^c||S|} \ln \frac{1}{\delta}}, \tag{12}
$$

*where* $v_1 = \rho^{L_W - 1} \varrho^{L_\Theta - 1} \|\tilde{\boldsymbol{X}}\|_F \sqrt{\ln m} \left(\prod_{l=1}^{L_W} a_l\right) \left(\sum_{l=1}^{L_W} \left(\frac{a_l'}{a_l}\right)^{2/3}\right)^{3/2} \left(\prod_{l=1}^{L_\Theta} b_l\right)$, $v_2 =$

$\sum_{l=1}^{L_\Theta} p_l p_{l-1}$, $v_3 = L_\Theta \rho^{L_W - 1} \varrho^{L_\Theta - 1} \mu^{-1} \|\tilde{\boldsymbol{X}}\|_F \left(\prod_{l=1}^{L_W} a_l\right) \left(\prod_{l=1}^{L_\Theta} b_l\right) \max_l \frac{b_l'}{b_l}$, *and* $C_1$, $C_2$ *are some absolute constants.*

First, let's show that the bound is non-trivial. Since activation functions are often at most 1-Lipschitz, we let $\rho = \varrho = 1$. Suppose $a_1 = \cdots = a_{L_W} = 1$ and $b_1 = \cdots = b_{L_\Theta} = 1$. Since $a_l'/a_l \leq \bar{d}_{l-1}$, we have $\left(\sum_{l=1}^{L_W} \left(a_l' a_l^{-1}\right)^{2/3}\right)^{3/2} \leq L_W^{3/2} \max_l a_l' a_l^{-1} \leq L_W^{3/2} \bar{d}$. In addition, $\max_l b_l'/b_l \leq \max_l \sqrt{p_l}$. Then the bound in Theorem 1 becomes

$$
\mathcal{L}_{S^c} \leq \mathcal{L}_S + \tilde{O}\left(\frac{\eta_\ell L_W^{3/2} \bar{d} \|\tilde{\boldsymbol{X}}\|_F}{|S|}\right) + \tilde{O}\left(\eta_\ell \mu \sqrt{\frac{\sum_{l=1}^{L_\Theta} p_l p_{l-1}}{|S|}}\right)
$$
$$
+ \frac{11\tau_\ell mn}{\sqrt{|S^c|}|S|} + 3\tau_\ell \sqrt{\frac{mn}{|S^c||S|} \ln \frac{1}{\delta}}. \tag{13}
$$

Note that $\|\tilde{\boldsymbol{X}}\|_F \leq \sqrt{mn} \max_{ij} |\hat{X}_{ij}|$. If $|S| > C_3 \max\left(L_W^{3/2} \bar{d}\sqrt{mn} \max_{ij} |\hat{X}_{ij}|, \sum_{l=1}^{L_\Theta} p_l p_{l-1}\right)$ holds for some constant $C_3$, the bound is non-trivial. Obviously, the condition holds if $n$ is much larger than $m$ and $|S|$ is sufficiently large. A smaller $\bar{d}$ leads to a tighter bound.

More specifically, Theorem 1 provides the following results.

**A. The error bound of prediction for the unknown ratings can be reduced via including the additional (element-wise) neural network.**

Given a fixed autoencoder, denote by $\mathcal{L}_S^0$ the training error without the element-wise neural network of NE-AECF. As discussed in Section 2, the additional neural network of NE-AECF aims to learn an activation function adaptively for the output layer of the decoder. Hence, the training error $\mathcal{L}_S^0$ can be reduced to $\mathcal{L}_S$. Denote $\mathcal{L}_S^0 - \mathcal{L}_S = \Delta_S$. In (13), $\varrho^{L_\Theta - 1}$, $\prod_{l=1}^{L_\Theta} b_l$, and $C_2 \eta_\ell \mu \sqrt{\frac{v_2 \ln v_3}{|S|}}$ are introduced by the additional neural network. Denote $v_1^0$ by the $v_1$ without $\varrho^{L_\Theta - 1}$ and $\prod_{l=1}^{L_\Theta} b_l$. Then we obtain $\mathcal{L}_{S^c}^0 \leq B_0$ while

$$
\mathcal{L}_{S^c} \leq B_0 \underbrace{- \Delta_S + (\varrho^{L_\Theta - 1} \prod_{l=1}^{L_\Theta} b_l - 1) \frac{C_1 \eta_\ell v_1^0 \ln |S|}{|S|} + C_2 \eta_\ell \mu \sqrt{\frac{v_2 \ln v_3}{|S|}}}_{\Delta}.
$$

Note that $\varrho^{L_\Theta - 1} \prod_{l=1}^{L_\Theta} b_l$ can be very close to 1 and $v_2$ is much smaller than $|S|$ (provided that the additional network is not too large). Therefore, $\Delta$ can be negative, which means the element-wise neural network can reduce the upper bound of the prediction error. On the other hand, if we do not use the additional neural network but still want reduce $B$ to $B_0 - \Delta_S$, we have to increase the depth or width of the main neural network, which will raise the value of $v_1$ and hence increase the upper bound. These results verified the superiority of our NE-AECF over classical autoencoder based CF methods such as the AutoRec of (Sedhain et al., 2015).

**B. Filling the unknown ratings with zeros reduces the upper bound of prediction error.**

In Theorem 1, if the unknown ratings are replaced by zeros, $\|\tilde{\boldsymbol{X}}\|_F$ will decrease. Hence the bound become tighter. Specifically, (13) becomes

$$
\begin{aligned}
\mathcal{L}_{S^c} \leq & \mathcal{L}_S + \tilde{O}\left(\frac{\eta_\ell L_W^{3/2} \bar{d} \max_{ij} |\tilde{X}_{ij}|}{\sqrt{|S|}}\right) + \tilde{O}\left(\eta_\ell \mu \sqrt{\frac{\sum_{l=1}^{L_\Theta} p_l p_{l-1}}{|S|}}\right) \\
& + \frac{11\tau_\ell mn}{\sqrt{|S^c||S|}} + 3\tau_\ell \sqrt{\frac{mn}{|S^c||S|} \ln \frac{1}{\delta}}.
\end{aligned}
\tag{14}
$$

The bound is tight if $\sqrt{|S|}$ is much larger than $\bar{d}$, the maximum size of the hidden layers.

**C. Increasing $n$ reduces the upper bound of prediction error.**

Let the sampling rate $\frac{|S|}{mn} \triangleq \zeta$ and network structures be fixed. Then (14) becomes

$$
\begin{aligned}
\mathcal{L}_{S^c} \leq & \mathcal{L}_S + \tilde{O}\left(\frac{\eta_\ell L_W^{3/2} \max_{ij} |\tilde{X}_{ij}|}{\sqrt{\zeta mn/\bar{d}^2}}\right) + \tilde{O}\left(\eta_\ell \mu \sqrt{\frac{\sum_{l=1}^{L_\Theta} p_l p_{l-1}}{\zeta mn}}\right) \\
& + \frac{11\tau_\ell}{\sqrt{(1-\zeta)\zeta^2 mn}} + 3\tau_\ell \sqrt{\frac{1}{(1-\zeta)\zeta mn} \ln \frac{1}{\delta}}.
\end{aligned}
\tag{15}
$$

Therefore, when $n$ increases, the bound becomes tighter. In addition, $\mathcal{L}_{S^c} \leq \mathcal{L}_S$ when $n \to \infty$. In real application, if there are more users than items, we need to construct a neural network such that the input is a vector of each user's rating, where items correspond to features and users correspond samples. In other word, larger difference between the number of users and the number of items leads to tighter upper bound of the prediction error, because we can construct the autoencoder along the smaller size of the rating matrix.

## 4 CONNECTION WITH PREVIOUS WORK

The element-wise neural network of NE-AECF can be regarded as an activation function adaptively learned from the data. It is closely related to the previous work on adaptive activation functions such (Lin et al., 2013; Agostinelli et al., 2014; Hou et al., 2017; Goyal et al., 2019). For instance, Lin et al. (2013) proposed to use micro neural networks to improve the convolution operator in convolutional neural networks. Hou et al. (2017) showed that applying adaptive activation functions in the regression (second-to-last) layer of a neural network can significantly decrease the bias of the regression. Their adaptive activation function is in the form of piece-wise polynomials. We found that, empirically, in NE-AECF, the improvement given by polynomials (8) and piece-wise polynomials (Hou et al., 2017) are not significant, which may be caused by the unboundedness of polynomials.

As mentioned in Introduction, theoretical study for autoencoder and deep learning based collaborative filtering is very limited. In recent years, a few researchers have studied the generalization ability or sample complexity of deep neural networks (Bartlett et al., 2017; Neyshabur et al., 2018; Golowich et al., 2018) but their results do not apply to autoencoder based CF and our AE-NECF. Our proof for Theorem 1 has taken advantage of the result of (Bartlett et al., 2017). There are two major differences. First, our setting is collaborative filtering, in which the training samples are matrix elements. Hence we utilized the transductive Rademacher complexity proposed by (El-Yaniv & Pechyony, 2009). Second, computing the complexity of the element-wise neural network required a different method rather than that of (Bartlett et al., 2017).

Shamir & Shalev-Shwartz (2014) provided the following generalization bound for nuclear norm minimization based CF: $\mathcal{L}_{S^c} \leq \mathcal{L}_S + O\left(\frac{\eta_\ell \|\hat{\boldsymbol{X}}\|_*(\sqrt{m}+\sqrt{n})}{|S|}\right) + R$, where $\hat{\boldsymbol{X}}$ denotes the recovered matrix given by nuclear norm minimization and $R$ stands for the remainder of their result (Theorem 6). Suppose the rank of $\hat{\boldsymbol{X}}$ is $\bar{d}$. Then the term related to the nuclear norm can be as large as $O\left(\frac{\eta_\ell \sqrt{\bar{d}} \|\hat{\boldsymbol{X}}\|_F (\sqrt{m}+\sqrt{n})}{\zeta mn}\right)$ or $O\left(\eta_\ell \mu \sqrt{\frac{\bar{d}}{\zeta^2 m}}\right)$, where we have assumed $m < n$. According to (15), the dominating term in our bound can be $\tilde{O}\left(\eta_\ell \mu \sqrt{\frac{\bar{d}}{\zeta^2 m}} \sqrt{\frac{L_W^3 \zeta \bar{d}}{n}}\right)$. Note that with the same $\bar{d}$, the

training error of our NE-AECF is less than the training error of nuclear norm minimization because we are using neural networks. Now we conclude that when $n$ is sufficiently large (compared to $L_W^3 \zeta \bar{d}$), our bound is tighter than that of (Shamir & Shalev-Shwartz, 2014).

## 5 NUMERICAL RESULTS

### 5.1 EXPERIMENTS ON MOVIELENS DATASETS

In this section, we evaluate the proposed method NE-AECF on two benchmark datasets of collaborative filtering: Movielens 100K and Movielens 1M (Harper & Konstan, 2015). These datasets contains 100 thousand (1 million) real world ratings for 1682 (3900) movies by 943 (6040) users, on a 5-stars scale. We randomly sample 90% of the known ratings as training set, leaving the remaining 10% as the test set. Among the training set, 5% are held out for hyperparameter tuning. Following Lee et al. (2016), test items or users without training observations are assigned with default rating of 3. The model performance is evaluated by the root mean squared error defined as

$$\text{RMSE} = \sqrt{\frac{\sum_{(i,j) \in S^c} (X_{ij} - \hat{X}_{ij})^2}{|S^c|}}, \tag{16}$$

where $S^c$ denotes the set of test ratings. For all the experiments, we use Adam (Kingma & Ba, 2015) to minimize the objective function of NE-AECF.

We report the mean RMSE based on 50 random splits and compare our NE-AECF with the following methods: BisaMF (Koren et al., 2009), NNMF (Dziugaite & Roy, 2015), AutoRec (Sedhain et al., 2015), CF-NADE (Zheng et al., 2016), LLOMRA (Lee et al., 2016), GC-MC (van den Berg et al., 2017), Sparse FC (Muller et al., 2018), DMF+ (Yi et al., 2019), and AutoRec W/SN (Yi et al., 2020). In our NE-AECF, the main neural network has one hidden layer. The numbers of hidden units are 700 and 2000 for MovieLens-100k and MovieLens-1M respectively. For both datasets, the element-wise neural network has one hidden layer, of which the size is 200. The regularization parameters $\lambda_W$ and $\lambda_\Theta$ were chosen from $\{0.01, 0.1, 1, 10, 50, 100, 200, 500\}$.

As shown[1] in Table 1, on MovieLens-100k, LLORMA outperformed all low-rank factorization methods, which indicates that a more sophisticated model (weighted sum of local low-rank models) fits the data better than a single low-rank model. As expected, our NE-AECF outperformed all baseline methods. Similarly, in Table 2, the RMSE of NE-AECF is less than those of other methods. It is worth mentioning that we have tried to increase the depth of the main neural network and the element-wise neural network, but the improvements in terms of RMSE are not significant.

Table 1: RMSE result of NE-AECF and compared methods on MovieLens-100k dataset.

| Model | ML-100k |
|---|---|
| BiasMF (Koren et al., 2009) | 0.911 |
| GC-MC (van den Berg et al., 2017) | 0.905 |
| AutoSVD++ (Zhang et al., 2017) | 0.904 |
| AutoSVD (Zhang et al., 2017) | 0.901 |
| NNMF (Dziugaite & Roy, 2015) | 0.903 |
| DMF+ (Yi et al., 2019) | 0.8889 |
| LLORMA (Lee et al., 2016) | 0.8881 |
| AutoRec w/SN (Yi et al., 2020) | 0.8816 |
| **NE-AECF** (ours) | **0.8791** $\pm$ 0.0063 |

### 5.2 EXPERIMENTS ON DOUBAN AND FLIXSTER

In this section, we evaluate the proposed NE-AECF on two more datasets Douban and Flixter, which are poplular benchmarks for recommendation systems. For these data, we use the preprocessed subsets and splits provided by Monti et al. (2017). These datasets both contain 3000 users and 3000

---

[1]The RMSEs of the compared methods shown in all tables are from the original papers, in which the experimental settings are the same or comparable.

Table 2: RMSE result of NE-AECF and compared methods on MovieLens-1M dataset

| Model | ML-1M |
|---|---|
| AutoSVD (Zhang et al., 2017) | 0.864 |
| AutoSVD++ (Zhang et al., 2017) | 0.848 |
| BiasMF (Koren et al., 2009) | 0.845 |
| NNMF (Dziugaite & Roy, 2015) | 0.843 |
| LLORMA (Lee et al., 2016) | 0.833 |
| DMF+ (Yi et al., 2019) | 0.8321 |
| GC-MC (van den Berg et al., 2017) | 0.832 |
| AutoRec (Sedhain et al., 2015) | 0.831 |
| CF-NADE (Zheng et al., 2016) | 0.829 |
| AutoRec w/SN (Yi et al., 2020) | 0.8260 |
| **NE-AECF** (ours) | $\mathbf{0.8252} \pm 0.0024$ |

items. Douban contains 136,891 ratings with density 0.0152 on a rating scale $\{1, 2, 3, 4, 5\}$. Flixster contains 26,173 ratings with density 0.0029 on a rating scale $\{0.5, 1, 1.5, ..., 5\}$, which is a bit different from the other three datasets. Among the training samples, 5% are used for hyperparameter tuning.

We report the mean RMSE on 5 repeated experiments and compare our NE-AECF with the following methods: PMF (Mnih & Salakhutdinov, 2008), GRALS (Rao et al., 2015), sRGCNN (Monti et al., 2017), GC-MC (van den Berg et al., 2017), Factorized EAE (Hartford et al., 2018) and GRAEM (Strahl et al., 2020). In our NE-AECF, the main neural network has one hidden layer, of which the size is 500, for both datasets. The structure of the element-wise neural network is the same as that used for the MovieLens datasets.

As shown in Table 3, our proposed NE-AECF outperforms other baseline method on both Douban and Flixster. Note that some of the compared methods include extra content like the side information of users and items into the model, while NE-AECF does not require extra content.

Table 3: RMSE result of NE-AECF and compared methods on Douban and Flixster dataset

| Model | Douban | Flixster |
|---|---|---|
| GRALS (Rao et al., 2015) | 0.8326 | 1.245 |
| PMF (Mnih & Salakhutdinov, 2008) | 0.7492 | 0.9809 |
| sRGCNN (Monti et al., 2017) | 0.801 | 0.926 |
| Factorized EAE (Hartford et al., 2018) | 0.738 | 0.908 |
| GC-MC (van den Berg et al., 2017) | 0.734 | 0.917 |
| GRAEM (Strahl et al., 2020) | 0.7497 | 0.8857 |
| **NE-AECF** (ours) | **0.7286** | **0.8816** |

## 6 CONCLUSION

This paper presented a novel collaborative filtering method called NE-AECF. We analyzed the generalization ability for NE-AECF and showed that the element-wise neural network is useful in reducing the upper bound of the prediction error in CF. The theoretical analysis indicates that filling the unknown ratings with zeros can make the error bound tighter. It also provides a guideline to make a choice between item-based autoencoder and user-based autoencoder. These theoretical findings were further validated by the numerical results, in which our NE-AECF has state-of-the-art performance in terms of RMSE.

It is possible to incorporate side information or graph neural network into our NE-AECF, which can be a future work.

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

## A  PROOF FOR THE MAIN THEOREM

First of all, we give the following lemmas.

**Lemma 1.** *Let* $\mathcal{H} = \{ \boldsymbol{H} \in \mathbb{R}^{m \times n} : h_{ij} = \boldsymbol{\Theta}_{L_\Theta} \sigma \left( \boldsymbol{\Theta}_{L_\Theta - 1} (\cdots \sigma(\boldsymbol{\Theta}_1 z_{ij}) \cdots) \right), \forall (i,j) \in [m] \times [n]; \boldsymbol{\Theta}_l \in \mathbb{R}^{p_l \times p_{l-1}}, \|\boldsymbol{\Theta}_l\|_2 \le b_l, \|\boldsymbol{\Theta}_l\|_F \le b'_l, \forall l \in [L_\Theta]; \boldsymbol{Z} \in \mathcal{Z}, \|\boldsymbol{Z}\|_F \le s_z \}, \text{ where the}$ *Lipschitz constant of* $\sigma$ *is* $\varrho$*. Suppose the covering number of* $\mathcal{Z}$ *with respect to* $\|\cdot\|_F$ *is upper-bounded by* $\kappa_\varepsilon$ *and* $\varepsilon = \epsilon \left( 2 \varrho^{L_\Theta - 1} \prod_{l=1}^{L_\Theta} b_l \right)^{-1}$*. Then the cover covering number of* $\mathcal{H}$ *with respect to* $\|\cdot\|_F$ *is bounded as*

$$\mathcal{N}(\mathcal{H}, \|\cdot\|_F, \epsilon) \le \kappa_\varepsilon \prod_{l=1}^{L_\Theta} \left( \frac{3\sqrt{2}\varrho^{L_\Theta-1}(L_\Theta+1)s_z \prod_{l=1}^{L_\Theta} b_l}{\epsilon} \right)^{p_l p_{l-1}}.$$

*Proof.* See Section B.1. □

Lemma 1 provides an upper bound of the covering number of the element-wise neural network. The following lemma shows an upper bound of the covering number of the main neural network.

**Lemma 2** (Theorem 3.3 of Bartlett et al. (2017), reformulated)**.** *Let* $\boldsymbol{Z} = \boldsymbol{W}_{L_W} \sigma \left( \boldsymbol{W}_{L_W-1} \left( \cdots \sigma(\boldsymbol{W}_1 \tilde{\boldsymbol{X}}) \cdots \right) \right)$*, where* $\boldsymbol{W}_l \in \mathbb{R}^{d_{l+1} d_l}$*,* $l \in [L_W]$*, and* $\max(m, d_1, \ldots, d_{L_W}) \le D$*. Denote the Lipschitz constant of* $\sigma$ *by* $\rho$*. Suppose the reference matrices* $(\boldsymbol{M}_1, \ldots, \boldsymbol{M}_{L_W})$ *are given. Define*

$$\mathcal{C} = \{ F_{\mathcal{W}}(\tilde{\boldsymbol{X}}) : \mathcal{W} = (\boldsymbol{W}_1, \ldots, \boldsymbol{W}_{L_W}), \|\boldsymbol{W}_l\|_\sigma \le a_l, \|\boldsymbol{W}_l - \boldsymbol{M}_l\|_\sigma \le a'_l, \forall \in [L_W] \}.$$

*The for any* $\epsilon > 0$*,*

$$\ln \mathcal{N}(\mathcal{C}, \epsilon, \|\cdot\|_F) \le \frac{\|\tilde{\boldsymbol{X}}\|_F^2 \ln 2D^2}{\epsilon^2} \left( \rho^{2(L_W-1)} \prod_{l=1}^{L_W} a_l^2 \right) \left( \sum_{l=1}^{L_W} \left( \frac{a'_l}{a_l} \right)^{2/3} \right)^3.$$

Now we can get an upper bound for the covering number of the entire neural network in NE-AECF.

**Lemma 3.** *The covering number of* $\mathcal{H}_{W,\Theta} = \{ H_\Theta \left( F_{\mathcal{W}(\tilde{\boldsymbol{X}})} \right) \}$ *with respect to* $\|\cdot\|_F$ *satisfies*

$$\ln \mathcal{N}(\mathcal{H}, \|\cdot\|_F, \epsilon) \le \frac{4\varrho^{2(L_\Theta-1)} \rho^{2(L_W-1)} \|\tilde{\boldsymbol{X}}\|_F^2 \ln 2D^2}{\epsilon^2} \left( \prod_{l=1}^{L_W} a_l^2 \right) \left( \prod_{l=1}^{L_\Theta} b_l^2 \right) \left( \sum_{l=1}^{L_W} \left( \frac{a'_l}{a_l} \right)^{2/3} \right)^3$$

$$+ \left( \sum_{l=1}^{L_\Theta} p_l p_{l-1} \right) \ln \left( \frac{6 L_\Theta \varrho^{L_\Theta-1} \rho^{L_W-1} \|\tilde{\boldsymbol{X}}\|_F \left( \prod_{l=1}^{L_W} a_l \right) \left( \prod_{l=1}^{L_\Theta} b_l \right) \max_l \frac{b'_l}{b_l}}{\epsilon} \right).$$

*Proof.* See Section B.2. □

Now we can calculate the upper bound of the Rademacher complexity of $\mathcal{H}_{W,\Theta}$ via using Lemma 3 and Dudley entropy integral bound.

**Lemma 4.** *Let* $v_1 = 4\rho^{2(L_W-1)}\varrho^{2(L_\Theta-1)}\|\tilde{\boldsymbol{X}}\|_F^2 \ln 2D^2 \left(\prod_{l=1}^{L_W} a_l^2\right) \left(\sum_{l=1}^{L_W} \left(\frac{a_l'}{a_l}\right)^{2/3}\right)^3 \left(\prod_{l=1}^{L_\Theta} b_l^2\right)$,

$v_2 = \sum_{l=1}^{L_\Theta} p_l p_{l-1}$, *and* $v_3 = 6L_\Theta\rho^{L_W-1}\varrho^{L_\Theta-1}\|\tilde{\boldsymbol{X}}\|_F \left(\prod_{l=1}^{L_W} a_l\right) \left(\prod_{l=1}^{L_\Theta} b_l\right) \max_l b_l'b_l^{-1}$. *Suppose* $\|h_\Theta(g_W(\tilde{\boldsymbol{X}}))\|_\infty \leq \mu$. *The Rademacher complexity of* $\mathcal{H}_{W,\Theta}$ *is bounded as*

$$\mathcal{R}_S(\mathcal{H}_{W,\Theta}) \leq \frac{4\mu}{S} + \frac{12\sqrt{v_1 + \mu^2 v_2}\ln S}{S} + \frac{12\mu\sqrt{v_2 \ln \mu^{-1}v_3}}{\sqrt{S}}. \tag{17}$$

*Proof.* See Section B.3. $\qquad\qquad\square$

The following lemma provides a sample complexity bound for transductive learning, which is consistent with the objective function and evaluation metric (RMSE) widely used in collaborative filtering.

**Lemma 5** (Corollary 1 of (El-Yaniv & Pechyony, 2009), reformulated)**.** *Let* $\mathcal{H}$ *be a fixed hypothesis set and suppose* $\sup_{i,j|\boldsymbol{X}\in\mathcal{H}} |\ell(Y_{ij}, X_{ij})| \leq \tau_\ell$. *Suppose a fixed set* $S$ *of distinct indices is uniformly and randomly split to two subsets* $S_{\text{train}}$ *and* $S_{\text{test}}$, *where*[2] $|S_{\text{test}}| > |S_{\text{train}}| > 50$. *Then with probability at least* $1 - \delta$ *over the random split, we have*

$$\frac{1}{|S_{\text{test}}|} \sum_{(i,j)\in S_{\text{test}}} \ell(Y_{ij}, X_{ij}) \leq \frac{1}{|S_{\text{train}}|} \sum_{(i,j)\in S_{\text{train}}} \ell(Y_{ij}, X_{ij}) + 4\mathcal{R}_S(\ell \circ \mathcal{H})$$
$$+ \frac{11\tau_\ell(|S_{\text{train}}| + |S_{\text{test}}|)}{\sqrt{|S_{\text{train}}||S_{\text{test}}|}} + 3\tau_\ell\sqrt{\frac{(|S_{\text{train}}| + |S_{\text{test}}|)}{|S_{\text{train}}||S_{\text{test}}|}\ln\frac{1}{\delta}} \tag{18}$$

Then Theorem 1 can be proved as follows.

*Proof.* Accoding to the Rademacher contraction property, we have $\mathcal{R}_S(\ell \circ \mathcal{H}) \leq \eta_\ell \mathcal{R}_S(\mathcal{H})$, where $\eta_\ell$ denotes the lipschitz constant of $\ell$. Using Lemma 5 with a slightly different notation and Lemma 4 where $\mu^2 v_2 \ll v_1$ provided that the element-wise neural network is small enough, we have

$$\frac{1}{|S^c|} \sum_{(i,j)\in S^c} \ell\left(X_{ij}, \hat{X}_{ij}\right) \leq \frac{1}{|S|} \sum_{(i,j)\in S} \ell\left(X_{ij}, \hat{X}_{ij}\right)$$
$$+ \eta_\ell\left(\frac{16\mu}{|S|} + \frac{48\sqrt{v_1 + \mu^2 v_2}\ln|S|}{|S|} + \frac{48\mu\sqrt{v_2 \ln \mu^{-1}v_3}}{\sqrt{|S|}}\right)$$
$$+ \frac{11\tau_\ell(|S| + |S^c|)}{\sqrt{|S||S^c|}} + 3\tau_\ell\sqrt{\frac{(|S| + |S^c|)}{|S||S^c|}\ln\frac{1}{\delta}}$$
$$\leq \frac{1}{|S|} \sum_{(i,j)\in S} \ell\left(X_{ij}, \hat{X}_{ij}\right)$$
$$+ \frac{C_1\eta_\ell v_1'\ln|S|}{|S|} + C_2\eta_\ell\mu\sqrt{\frac{v_2 \ln v_3'}{|S|}}$$
$$+ \frac{11\tau_\ell mn}{\sqrt{|S||S^c|}} + 3\tau_\ell\sqrt{\frac{mn}{|S||S^c|}\ln\frac{1}{\delta}}, \tag{19}$$

where $C_1$ and $C_2$ are some fixed constants, and

$$v_1' = \rho^{(L_W-1)}\varrho^{(L_\Theta-1)}\|\boldsymbol{X}\|_F\sqrt{\ln D}\left(\prod_{l=1}^{L_W} a_l\right)\left(\sum_{l=1}^{L_W}\left(\frac{a_l'}{a_l}\right)^{2/3}\right)^{3/2}\left(\prod_{l=1}^{L_\Theta} b_l\right),$$

---

[2]We use these assumptions to simplify the theorem.

and $v_3' = L_\Theta \mu^{-1} \gamma \rho^{L_W - 1} \varrho^{L_\Theta - 1} \|\boldsymbol{X}\|_F \left( \prod_{l=1}^{L_W} a_l \right) \left( \prod_{l=1}^{L_\Theta} b_l \right)$. Rename $v_1'$ and $v_3'$ as $v_1$ and $v_3$ respectively, we finish the proof. □

## B  PROOF FOR LEMMAS

### B.1  PROOF FOR LEMMA 1

*Proof.* Let $\mathcal{S}_{\Theta_l} := \{\boldsymbol{\Theta}_l \in \mathbb{R}^{p_{l+1} \times p_l} : \|\boldsymbol{\Theta}_l\|_2 \le b_l, \|\boldsymbol{\Theta}_l\|_F \le b_l'\}, \forall l \in [L_\Theta]$. It is known that there exists an $\epsilon_l$-net $\bar{\mathcal{S}}_{\Theta_l}$ obeying

$$\mathcal{N}(\mathcal{S}_{\Theta_l}, \|\cdot\|_F, \epsilon_l) \le \left( \frac{3b_l'}{\epsilon_l} \right)^{p_l p_{l-1}}$$

such that $\|\boldsymbol{\Theta}_l - \bar{\boldsymbol{\Theta}}_l\|_F \le \epsilon_l$. We have

$$
\begin{aligned}
&|h_{ij} - \bar{h}_{ij}| = \|h_{ij} - \bar{h}_{ij}\|_F \\
&= \left\| \boldsymbol{\Theta}_{L_\Theta} \sigma\left(\boldsymbol{\Theta}_{L_\Theta - 1}(\cdots \sigma(\boldsymbol{\Theta}_1 z_{ij})\cdots)\right) - \bar{\boldsymbol{\Theta}}_{L_\Theta} \sigma\left(\bar{\boldsymbol{\Theta}}_{L_\Theta - 1}(\cdots \sigma(\bar{\boldsymbol{\Theta}}_1 \bar{z}_{ij})\cdots)\right) \right\|_F \\
&= \Big\| \boldsymbol{\Theta}_{L_\Theta} \sigma\left(\boldsymbol{\Theta}_{L_\Theta - 1}(\cdots \sigma(\boldsymbol{\Theta}_1 z_{ij})\cdots)\right) - \bar{\boldsymbol{\Theta}}_{L_\Theta} \sigma\left(\boldsymbol{\Theta}_{L_\Theta - 1}(\cdots \sigma(\boldsymbol{\Theta}_1 z_{ij})\cdots)\right) \\
&\quad + \bar{\boldsymbol{\Theta}}_{L_\Theta} \sigma\left(\boldsymbol{\Theta}_{L_\Theta - 1}(\cdots \sigma(\boldsymbol{\Theta}_1 z_{ij})\cdots)\right) - \bar{\boldsymbol{\Theta}}_{L_\Theta} \sigma\left(\bar{\boldsymbol{\Theta}}_{L_\Theta - 1}(\cdots \sigma(\boldsymbol{\Theta}_1 z_{ij})\cdots)\right) + \cdots \\
&\quad + \bar{\boldsymbol{\Theta}}_{L_\Theta} \sigma\left(\bar{\boldsymbol{\Theta}}_{L_\Theta - 1}(\cdots \sigma(\boldsymbol{\Theta}_1 z_{ij})\cdots)\right) - \bar{\boldsymbol{\Theta}}_{L_\Theta} \sigma\left(\bar{\boldsymbol{\Theta}}_{L_\Theta - 1}(\cdots \sigma(\bar{\boldsymbol{\Theta}}_1 z_{ij})\cdots)\right) \Big\|_F \\
&\quad + \bar{\boldsymbol{\Theta}}_{L_\Theta} \sigma\left(\bar{\boldsymbol{\Theta}}_{L_\Theta - 1}(\cdots \sigma(\boldsymbol{\Theta}_1 z_{ij})\cdots)\right) - \bar{\boldsymbol{\Theta}}_{L_\Theta} \sigma\left(\bar{\boldsymbol{\Theta}}_{L_\Theta - 1}(\cdots \sigma(\bar{\boldsymbol{\Theta}}_1 \bar{z}_{ij})\cdots)\right) \Big\|_F \\
&\le \left\| \boldsymbol{\Theta}_{L_\Theta} \sigma\left(\boldsymbol{\Theta}_{L_\Theta - 1}(\cdots \sigma(\boldsymbol{\Theta}_1 z_{ij})\cdots)\right) - \bar{\boldsymbol{\Theta}}_{L_\Theta} \sigma\left(\boldsymbol{\Theta}_{L_\Theta - 1}(\cdots \sigma(\boldsymbol{\Theta}_1 z_{ij})\cdots)\right) \right\|_F \\
&\quad + \left\| \bar{\boldsymbol{\Theta}}_{L_\Theta} \sigma\left(\boldsymbol{\Theta}_{L_\Theta - 1}(\cdots \sigma(\boldsymbol{\Theta}_1 z_{ij})\cdots)\right) - \bar{\boldsymbol{\Theta}}_{L_\Theta} \sigma\left(\bar{\boldsymbol{\Theta}}_{L_\Theta - 1}(\cdots \sigma(\boldsymbol{\Theta}_1 z_{ij})\cdots)\right) \right\|_F + \cdots \\
&\quad + \left\| \bar{\boldsymbol{\Theta}}_{L_\Theta} \sigma\left(\bar{\boldsymbol{\Theta}}_{L_\Theta - 1}(\cdots \sigma(\boldsymbol{\Theta}_1 z_{ij})\cdots)\right) - \bar{\boldsymbol{\Theta}}_L \sigma\left(\bar{\boldsymbol{\Theta}}_{L_\Theta - 1}(\cdots \sigma(\bar{\boldsymbol{\Theta}}_1 z_{ij})\cdots)\right) \right\|_F \\
&\quad + \left\| \bar{\boldsymbol{\Theta}}_{L_\Theta} \sigma\left(\bar{\boldsymbol{\Theta}}_{L_\Theta - 1}(\cdots \sigma(\bar{\boldsymbol{\Theta}}_1 z_{ij})\cdots)\right) - \bar{\boldsymbol{\Theta}}_{L_\Theta} \sigma\left(\bar{\boldsymbol{\Theta}}_{L_\Theta - 1}(\cdots \sigma(\bar{\boldsymbol{\Theta}}_1 \bar{z}_{ij})\cdots)\right) \right\|_F \\
&\overset{(a)}{\le} \varrho^{L_\Theta - 1} |z_{ij}| \left\| \boldsymbol{\Theta}_{L_\Theta} - \bar{\boldsymbol{\Theta}}_{L_\Theta} \right\|_F \prod_{l=1}^{L_\Theta - 1} \|\boldsymbol{\Theta}_l\|_2 \\
&\quad + \varrho^{L_\Theta - 1} |z_{ij}| \left\| \bar{\boldsymbol{\Theta}}_{L_\Theta} \right\|_2 \left\| \boldsymbol{\Theta}_{L_\Theta - 1} - \bar{\boldsymbol{\Theta}}_{L_\Theta - 1} \right\|_F \prod_{l=1}^{L_\Theta - 2} \|\boldsymbol{\Theta}_l\|_2 + \cdots \\
&\quad + \varrho^{L-1} |z_{ij}| \left( \prod_{l=2}^{L_\Theta} \left\| \bar{\boldsymbol{\Theta}}_l \right\|_2 \right) \left\| \boldsymbol{\Theta}_1 - \bar{\boldsymbol{\Theta}}_1 \right\|_F \\
&\quad + \varrho^{L_\Theta - 1} \|z_{ij} - \bar{z}_{ij}\|_F \prod_{l=1}^{L_\Theta} \|\bar{\boldsymbol{\Theta}}_l\|_2 \\
&\le \varrho^{L_\Theta - 1} \left( s_{z_{ij}} \epsilon_{L_\Theta} \prod_{l \ne L_\Theta} b_l + s_{z_{ij}} \epsilon_{L_\Theta - 1} \prod_{l \ne L_\Theta - 1} b_l + s_{z_{ij}} \epsilon_1 \prod_{l \ne 1}^{L_\Theta} b_l + \cdots + \|z_{ij} - \bar{z}_{ij}\|_F \prod_{l=1}^{L_\Theta} b_l \right).
\end{aligned}
$$
(20)

In (a), we used the facts $\|XY\|_F \leq \|X\|_2\|Y\|_F$ and $\|\sigma(X)-\sigma(Y)\|_F \leq \varrho\|X-Y\|_F$ recursively. It follows that

$$
\|H - \bar{H}\|_F = \sqrt{\sum_{ij} h_{ij}^2}
$$

$$
\overset{(a)}{\leq} \varrho^{L_\Theta-1} \sqrt{\sum_{ij} 2\left( s_{z_{ij}}^2 \left( \epsilon_{L_\Theta} \prod_{l\neq L_\Theta} b_l + \epsilon_{L_\Theta-1} \prod_{l\neq L_\Theta-1} b_l + \cdots + \epsilon_1 \prod_{l\neq 1}^{L_\Theta} b_l \right)^2 + \|z_{ij} - \hat{z}_{ij}\|_F^2 \left( \prod_{l=1}^{L_\Theta} s_l \right)^2 \right)}
$$

$$
= \sqrt{2}\varrho^{L_\Theta-1} \sqrt{\left( \epsilon_{L_\Theta} \prod_{l\neq L_\Theta} b_l + \epsilon_{L_\Theta-1} \prod_{l\neq L_\Theta-1} b_l + \cdots + \epsilon_1 \prod_{l\neq 1}^{L_\Theta} b_l \right)^2 \|Z\|_F^2 + \left( \prod_{l=1}^{L_\Theta} b_l \right)^2 \|Z - \bar{Z}\|_F^2}
$$

$$
\leq \sqrt{2}\varrho^{L_\Theta-1} \sqrt{\left( \epsilon_{L_\Theta} \prod_{l\neq L_\Theta} b_l + \epsilon_{L_\Theta-1} \prod_{l\neq L_\Theta-1} b_l + \cdots + \epsilon_1 \prod_{l\neq 1}^{L_\Theta} b_l \right)^2 s_z^2 + \left( \prod_{l=1}^{L_\Theta} b_l \right)^2 \epsilon_z^2}.
$$

$$(21)$$

In (a), we used the fact $(x + y)^2 \leq 2(x^2 + y^2)$. Let $\epsilon_l = \dfrac{\epsilon/(\sqrt{2}L_\Theta)}{\sqrt{2}\varrho^{L_\Theta-1}s_z \prod_{k\neq l} b_k}$, $\forall l \in [L_\Theta]$. Let $\epsilon_z = \dfrac{\epsilon/\sqrt{2}}{\sqrt{2}\varrho^{L_\Theta-1} \prod_{l=1}^{L_\Theta} b_l}$. We arrive at

$$
\|H - \bar{H}\|_F \leq \epsilon. \tag{22}
$$

It means that $\bar{H}$ is an $\epsilon$-cover of $H$. Then the covering number of $\mathcal{H}$ is bounded as

$$
\mathcal{N}(\mathcal{H}, \|\cdot\|_F, \epsilon)
$$

$$
\leq \mathcal{N}(\mathcal{Z}, \|\cdot\|_F, \epsilon_z) \prod_{l=1}^{L_\Theta} \mathcal{N}(\mathcal{S}_{\Theta_l}, \|\cdot\|_F, \epsilon_l)
$$

$$
\leq \kappa_\varepsilon \prod_{l=1}^{L_\Theta} \left( \frac{6\varrho^{L_\Theta-1}L_\Theta s_z b_l' \prod_{k\neq l} b_k}{\epsilon} \right)^{p_l p_{l-1}} \tag{23}
$$

$$
= \kappa_\varepsilon \prod_{l=1}^{L_\Theta} \left( \frac{6\varrho^{L_\Theta-1}L_\Theta s_z b_l' b_l^{-1} \prod_{k=1}^{L_\Theta} b_k}{\epsilon} \right)^{p_l p_{l-1}}
$$

$$
\leq \kappa_\varepsilon \left( \frac{C_\Theta}{\epsilon} \right)^{\sum_{l=1}^{L_\Theta} p_l p_{l-1}},
$$

where $C_\Theta = 6\varrho^{L_\Theta-1}L_\Theta s_z \gamma \prod_{l=1}^{L_\Theta} b_l$ and $\gamma = \max_l b_l' b_l^{-1}$. This finished the proof. $\square$

## B.2 Proof for Lemma 3

*Proof.* It is easy to show that

$$
s_z \geq \|Z\|_F = \left\| W_{L_W}\left( \sigma_W\left( W_{L_W-1}\sigma_W(\cdots\sigma_W(W_1\tilde{X})\cdots) \right) \right) \right\|_F \tag{24}
$$

$$
\geq \rho^{L_W-1}\|X\|_F \prod_{l=1}^{L_W} a_l. \tag{25}
$$

Combining Lemma 1 and Lemma 2, we have

$$
\ln \mathcal{N}(\mathcal{H}, \|\cdot\|_F, \epsilon)
$$

$$
\leq \ln \kappa_\varepsilon + \left( \sum_{l=1}^{L_\Theta} p_l p_{l-1} \right) \ln(\frac{C_\Theta}{\epsilon})
$$

$$
\leq \frac{4\rho^{2(L_W-1)} \varrho^{2(L_\Theta-1)} \|\boldsymbol{X}\|_F^2 \ln 2D^2}{\epsilon^2} \left( \prod_{l=1}^{L_W} a_l^2 \right) \left( \sum_{l=1}^{L_W} \left( \frac{a_l'}{a_l} \right)^{2/3} \right)^3 \left( \prod_{l=1}^{L_\Theta} b_l^2 \right) \tag{26}
$$

$$
+ \left( \sum_{l=1}^{L_\Theta} p_l p_{l-1} \right) \ln \left( \frac{6 L_\Theta \varrho^{L_\Theta-1} \rho^{L_W-1} \|\boldsymbol{X}\|_F \left( \prod_{l=1}^{L_W} a_l \right) \left( \prod_{i=1}^{L_\Theta} b_l \right) \max_l \frac{b_l'}{b_l}}{\epsilon} \right).
$$

$\square$

### B.3 PROOF FOR LEMMA 4

Before proof, we give the following lemma, which is a variant of the Dudley entropy integral bound on Rademacher complexity.

**Lemma 6** (Lemma A.5 of Bartlett et al. (2017)). *Let $\mathcal{F}$ be a real-valued function class taking values in $[0, 1]$ and assume that $\boldsymbol{0} \in \mathcal{F}$. Then*

$$
\mathcal{R}(\mathcal{F}_{|S}) \leq \inf_{\alpha>0} \left( \frac{4\alpha}{\sqrt{S}} + \frac{12}{S} \int_\alpha^{\sqrt{S}} \sqrt{\ln \mathcal{S}(\mathcal{F}_{|S}, \|\cdot\|_F, \epsilon)} d\epsilon \right). \tag{27}
$$

*Proof.* For convenience, let

$$
v_1 = 4\rho^{2(L_W-1)} \varrho^{2(L_\Theta-1)} \|\boldsymbol{X}\|_F^2 \ln 2D^2 \left( \prod_{l=1}^{L_W} a_l^2 \right) \left( \sum_{l=1}^{L_W} \left( \frac{a_l'}{a_l} \right)^{2/3} \right)^3 \left( \prod_{l=1}^{L_\Theta} b_l^2 \right),
$$

$v_2 = \sum_{l=1}^{L_\Theta} p_l p_{l-1}$, and $v_3 = 6 L_\Theta \rho^{L_W-1} \varrho^{L_\Theta-1} \|\boldsymbol{X}\|_F \left( \prod_{l=1}^{L_W} a_l \right) \left( \prod_{l=1}^{L_\Theta} b_l \right) \max_l b_l' b_l^{-1}$. Then $\ln \mathcal{N}(\mathcal{H}, \|\cdot\|_F, \epsilon) \leq \frac{v_1}{\epsilon^2} + v_2 \ln \left( \frac{v_3}{\epsilon} \right)$ from (26). Let $\mu = \max_{H \in \mathcal{H}} \|H\|_\infty$ and $\bar{\mathcal{H}} = \{\bar{H} : \mu \bar{H} \in \mathcal{H}\}$. It follows that $\ln \mathcal{N}(\bar{\mathcal{H}}, \|\cdot\|_F, \epsilon) \leq \frac{\mu^{-2} v_1}{\epsilon^2} + v_2 \ln \left( \frac{\mu^{-1} v_3}{\epsilon} \right)$.

According to Lemma 6, we have

$$
\begin{aligned}
\mathcal{R}_S(\bar{\mathcal{H}}) &\leq \inf_{\alpha>0} \left( \frac{4\alpha}{\sqrt{S}} + \frac{12}{S} \int_\alpha^{\sqrt{S}} \sqrt{\frac{\mu^{-2} v_1}{\epsilon^2} + v_2 \ln \left( \frac{\mu^{-1} v_3}{\epsilon} \right)} d\epsilon \right) \\
&\overset{(a)}{\leq} \inf_{\alpha>0} \left( \frac{4\alpha}{\sqrt{S}} + \frac{12}{S} \int_\alpha^{\sqrt{S}} \sqrt{\frac{\mu^{-2} v_1 + v_2}{\epsilon^2} + v_2 \ln \mu^{-1} v_3} d\epsilon \right) \\
&\overset{(b)}{\leq} \inf_{\alpha>0} \left( \frac{4\alpha}{\sqrt{S}} + \frac{12}{S} \int_\alpha^{\sqrt{S}} \left( \frac{\sqrt{\mu^{-2} v_1 + v_2}}{\epsilon} + \sqrt{v_2 \ln \mu^{-1} v_3} \right) d\epsilon \right) \\
&= \inf_{\alpha>0} \left( \frac{4\alpha}{\sqrt{S}} + \frac{12}{S} \left( \sqrt{\mu^{-2} v_1 + v_2} \ln \frac{\sqrt{S}}{\alpha} + \sqrt{v_2 \ln \mu^{-1} v_3} \left( \sqrt{S} - \alpha \right) \right) \right) \\
&\overset{(c)}{\leq} \frac{4}{S} + \frac{12 \sqrt{\mu^{-2} v_1 + v_2}}{S} \ln S + \frac{12(S-1) \sqrt{v_2 \ln \mu^{-1} v_3}}{S \sqrt{S}} \\
&\leq \frac{4}{S} + \frac{12 \sqrt{\mu^{-2} v_1 + v_2} \ln S}{S} + \frac{12 \sqrt{v_2 \ln \mu^{-1} v_3}}{\sqrt{S}}.
\end{aligned} \tag{28}
$$

In (a), we used the fact $\ln \frac{1}{x} \leq \frac{1}{x^2}$. The inequality (b) holds according to $\sqrt{x+y} \leq \sqrt{x} + \sqrt{y}$. In (c), we have let $\alpha = \frac{1}{\sqrt{S}}$, which though may not be the best choice. We arrive at

$$\mathcal{R}_S(\mathcal{H}_{W,\Theta}) \leq \frac{4\mu}{S} + \frac{12\sqrt{v_1 + \mu^2 v_2}\ln S}{S} + \frac{12\mu\sqrt{v_2 \ln \mu^{-1} v_3}}{\sqrt{S}}.$$

$\square$

