# OpenReview forum: "Neuron-Enhanced Autoencoder based Collaborative filtering: Theory and Practice"
_ICLR.cc/2022/Conference — ICLR 2022 Submitted_

### Official Review · Reviewer_83pQ · 2021-10-22

**Correctness:** 1
**Technical Novelty And Significance:** 2
**Empirical Novelty And Significance:** 1
**Recommendation:** 1
**Confidence:** 5

**Main Review:**

Evaluation based on RMSE on the OBSERVED ratings in experiments on recommender systems triggers an immediate reject of the paper in my view.

The reason is that this implicitly assumes that the data are missing at random, while in fact they are missing not at random (MNAR), which is one of the core properties of the recommendation problem (and  which makes it different from matrix completion and compressive sensing). This fact was realized in the literature a decade ago (Merlin et al, 2009, RecSys conference best paper), which has led the community away from using RMSE on the OBSERVED ratings, and towards using metrics that take into account ALL user-item-interactions, whether observed or not. Taking into account ALL user-item interactions can most easily be accomplished with ranking mertics, which have since been used in the literature. But it can also be done in the context of rating prediction, which however becomes a much more complicated problem in this case. Some references on the data being MNAR and ranking metrics:
- B. Marlin and R. Zemel: Collaborative Prediction and Ranking with Non-Random Missing Data, Recsys 2009
- H. Steck: Training and testing of recommender systems on data missing not at random, KDD 2010

And some references on rating prediction done right, i.e., accounting for the data being MNAR:
- J.M. Hernández-Lobato, N. Houlsby, and Z. Ghahramani. Probabilistic matrix factorization with non-random missing data. ICML 2014.
- D. Liang, L. Charlin, J. McInerney, D. M. Blei. Modeling User Exposure in Recommendation. WWW 2016.

Apart from that, two  rather small data sets are used in the experiments (Movielens 100k and 1 million), which raises the question of scalability of the proposed approach.


**Summary Of The Paper:**

This paper considers an autoencoder with one hidden layer. While the output layer uses a linear activation function in several papers in the literature, this paper proposes to use a nonlinear activation function in the output layer. Moreover, instead of using a fixed function (e.g., sigmoid), this non-linear activation function is learned in this paper, using a parametrized form. Generalization bounds are  derived, and experiments are conducted.


**Summary Of The Review:**

Evaluation based on RMSE on the OBSERVED ratings in experiments on recommender systems triggers an immediate reject of the paper in my view.

---

### Official Review · Reviewer_rgcR · 2021-10-31

**Correctness:** 3
**Technical Novelty And Significance:** 2
**Empirical Novelty And Significance:** 2
**Recommendation:** 3
**Confidence:** 3

**Main Review:**

Strengths
1. The proposed method is easy to implement.
2. A detailed theoretical analysis is provided to prove the effectiveness of the element-wise neural network.
3. This paper draws some interesting conclusions like, the data sparsity is useful.

Weaknesses
1. The assumptions in theorem 1 may be rough and untenable. For example, the authors suppose that the ratings are observed uniformly and randomly. However, the missing ratings in recommender systems are usually not missing at random. Multiple types of biases have impact on the observable ratings. I am not sure if the conclusions of this paper hold when the ratings are not uniformly observed.
2. The authors investigate the rating prediction problem in recommender systems. However, most real-world recommender systems pay more attention to the item recommendation problem concerning the ranks of items and using different losses such as cross-entropy.
3. The experiments are insufficient.  Although they test the proposed method on four datasets, they only conduct experiments for performance comparison. In two subsections, they evaluate the method on movielens-10k (1M), and Douban and Flixster, respectively. I think there is no need to separate the performance comparison into two subsections while providing nothing new. The additional two datasets are also small, and the experimental results are therefore less convincing. Since the proposed method has a very simple structure, it is meaningful to investigate if the proposed method shows the same advantages when using different nonlinear activation functions, neuron sizes and layers. Besides, the experimental results are not significant.
4. The writing is not concise. I think there is no need to provide too much basic knowledge in section 2. Instead, more experiments are needed.


**Summary Of The Paper:**

This paper proposes to use an additional element-wise neural network to enhance the reconstruction ability of the autoencoder for recommendation. A detailed theoretical analysis is provided that proves the element-wise neural network is able to reduce the upper bound of the prediction error for the unknown ratings.

**Summary Of The Review:**

This paper proposes to use an additional element-wise neural network to enhance the autoencoder for recommendation, and provides detailed theoretical analysis. However, the authors do not comprehensively evaluate their model and only very few experiments are conducted. The fundamental assumption of the proposed method is problematic and the experimental results only show slight improvements compared with existing method.

---

### Official Review · Reviewer_7kpq · 2021-11-01

**Correctness:** 4
**Technical Novelty And Significance:** 3
**Empirical Novelty And Significance:** 2
**Recommendation:** 5
**Confidence:** 4

**Main Review:**

Major concerns:

1. The theoretical contribution is not easy and intuitive to follow. Moreover, I am not sure how significant or generalizable the theoretical contribution is.

2. The empirical results are not solid enough. For example, in Table 1, the proposed method seems to have achieved the best performance, while CF-UIcA w/ SN yielded lower RMSE according to the SN paper. Also, the experiment of ML 10M is missing.

3. SN is simple and boosts most algorithms; it will be nice to present the proposed idea similarly, i.e., boosting most existing algorithms, such as CF-UIcA. There is no obvious difficulty in not doing that. Please correct me if I am wrong.

4. Could we add one more non-linear layer between hidden and output?

I will increase the score if there are more promising empirical results.


**Summary Of The Paper:**

This paper proposes an imputation method that consists of two networks. One network is AE, and one network is element-wise non-linear transformation. The authors proved the proposed method has a better generalization bound for unknown ratings. Furthermore, the authors demonstrated the proposed method achieved promising results in a few datasets.


**Summary Of The Review:**

This work proposes imputing the missingness through two NNs, addressing the non-linear transformation between hidden and output. The authors provided theoretical justification for why the proposed method is better, but the empirical experiments can be improved.

---

### Official Review · Reviewer_WENm · 2021-11-02

**Correctness:** 3
**Technical Novelty And Significance:** 3
**Empirical Novelty And Significance:** 3
**Recommendation:** 5
**Confidence:** 4

**Main Review:**

Strengths:

1 The idea of introducing an elementwise function (approximated by a neural network) is new.

2 The results show that the newly introduced neural network is helpful in improving the prediction accuracy.

Weaknesses:

1 Collaborative rating prediction is a very well-studied problem, for which there are lots of existing works. Moreover, in most real recommender systems, item ranking is more consistent with a real setting.

2 The time complexity seems rather high. First, the authors use an item-oriented autoencoder, in which there may be lots of users associated with a typical item. Second, the elementwise function is expensive. Third, the number of hidden units is much larger than a typical matrix factorization-based method.

3 The authors do not provide sufficient details or justification on using a large number of hidden units and an additional elementwise function. Moreover, treating unobserved ratings as zeros may introduce bias, which is also not justified.


**Summary Of The Paper:**

In this paper, the authors study a traditional collaborative filtering problem with users' ratings, where the goal is to predict ratings that are unobserved. Specifically, the authors propose an enhanced autoencoder-based method called NE-AECF. The main idea of NE-AECF as shown in Eq(10) and Figure 1 is that it contains an additional module for each predicted rating by an autoencoder, i.e., h_\theta(g_w) in Eq(10) and the most right part near the output layer in Figure 1.

**Summary Of The Review:**

In this paper, the authors study a traditional collaborative filtering problem with users' ratings, where the goal is to predict ratings that are unobserved. Specifically, the authors propose an enhanced autoencoder-based method called NE-AECF. The main idea of NE-AECF as shown in Eq(10) and Figure 1 is that it contains an additional module for each predicted rating by an autoencoder, i.e., h_\theta(g_w) in Eq(10) and the most right part near the output layer in Figure 1.

The authors conduct experiments on some public datasets and show that the newly introduced neural network is helpful in improving the prediction accuracy.

Some comments:

1 Collaborative rating prediction is a very well-studied problem, for which there are lots of existing works. Moreover, in real recommender systems, item ranking is more consistent with a real setting.

2 The time complexity seems rather high. First, the authors use an item-oriented autoencoder, in which there may be lots of users associated with a typical item. Second, the elementwise function is expensive. Third, the number of hidden units is much larger than a typical matrix factorization-based method.

3 The authors do not provide sufficient details or justification on using a large number of hidden units and an additional elementwise function. Moreover, treating unobserved ratings as zeros may introduce bias, which is also not justified.

---

### Decision · Program_Chairs · 2022-01-20

**Decision:**

Reject

**Comment:**

This paper proposed an enhanced autoencoder for collaborative filtering by adding another element-wise neural network for rating predictions. Overall the scores are negative, where reviewers pointed out concerns around the motivation, time complexities, and most importantly, using rating prediction as the evaluation setting which ignores the missing-not-at-random nature of the recommender systems. The authors didn't provide any response. Therefore, I vote for rejection.